# Degradation of Three Microcystin Variants in the Presence of the Macrophyte *Spirodela polyrhiza* and the Associated Microbial Communities

**DOI:** 10.3390/ijerph19106086

**Published:** 2022-05-17

**Authors:** Magdalena Toporowska

**Affiliations:** Department of Hydrobiology and Protection of Ecosystems, University of Life Sciences in Lublin, Dobrzańskiego 37, 20-262 Lublin, Poland; magdalena.toporowska@up.lublin.pl; Tel.: +48-81-461-00-61 (ext. 309)

**Keywords:** cyanotoxins, microcystins, biodegradation, bacteria, algae, MC bioaccumulation

## Abstract

Cyanobacteria, which form water blooms all over the world, can produce a wide range of cyanotoxins such as hepatotoxic microcystins (MCs) and other biologically active metabolites harmful to living organisms, including humans. Microcystin biodegradation, particularly caused by bacteria, has been broadly documented; however, studies in this field focus mainly on strains isolated from natural aquatic environments. In this paper, the biodegradation of microcystin-RR (MC-RR), microcystin-LR (MC-LR), and microcystin-LF (MC-LF) after incubation with *Spirodela polyrhiza* and the associated microorganisms (which were cultured under laboratory conditions) is shown. The strongest MC biodegradation rate after nine days of incubation was observed for MC-RR, followed by MC-LR. No statistically significant decrease in the concentration of MC-LF was noted. Products of MC decomposition were detected via the HPLC method, and their highest number was found for MC-RR (six products with the retention time between 5.6 and 16.2 min), followed by MC-LR (two products with the retention time between 19.3 and 20.2 min). Although the decrease in MC-LF concentration was not significant, four MC-LF degradation products were detected with the retention time between 28.9 and 33.0 min. The results showed that MC-LF was the most stable and resistant MC variant under experimental conditions. No accumulation of MCs or their biodegradation products in *S. polyrhiza* was found. The findings suggest that the microorganisms (bacteria and algae) associated with *S. polyrhiza* could be responsible for the MC biodegradation observed. Therefore, there is a need to broaden the research on the biodegradation products detected and potential MC-degraders associated with plants.

## 1. Introduction

Cyanobacterial blooms (HCBs) and cyanotoxin production are growing global problems connected with climate changes and water eutrophication [1,2,3,4]. Microcystins (MCs) are the most commonly occurring and studied toxins produced by cyanobacteria [4,5]. Over 240 MC variants have been characterized [6], out of which microcystin-LR (MC-LR), microcystin-RR (MC-RR), and microcystin-YR (MC-YR) are very toxic, the most frequently occurring in freshwater bodies, and extensively studied isoforms. Microcystin-LF (MC-LF) and microcystin-LW (MC-LW) have also been found in high concentrations in eutrophicated waters ([7,8] and references therein). In general, MCs are monocyclic heptapeptides (cyclo-(-D-Ala-L-X-D-MeAsp-L-Z-Adda-D-Glu-Mdha)) which exert severe negative effects on microorganisms, plants, animals, aquatic ecosystems, and humans [9,10,11,12]. The unique structure of Adda is crucial for the biological activity of MCs [13]. MCs mainly differ in amino acids and methylation or demethylation on MeAsp and Mdha. For example, MC-LR contains leucine (L) and arginine (R) amino acids, MC-RR consists of two ‘R’ amino acids, and MC-LF contains leucine (L) and phenylalanine (F) amino acids. As many surface waters are used as a source of drinking water, food, or for recreation, the health problems related to MC toxicity are well-documented [8,9,14]. Cyanobacteria, which proliferate commonly in eutrophic waters and can produce MCs, belong to multiple taxa within the genera *Microcystis*, *Aphanizomenon*, *Dolichospermum*, *Anabaena*, *Planktothrix*, *Oscillatoria*, *Nostoc* [13,15,16,17].

Due to the common occurrence of MCs and their threat to aquatic ecosystems and human health, data on possible MC degradation processes and products (which regulate the ambient concentration of these toxins) is of crucial significance [18,19,20]. MCs are resistant to different physical and chemical factors, including extremely low and high temperatures or pH [21]. Organisms able to biodegrade MCs belong to aerobic and anaerobic bacteria, fungi, zooplankton, and plants [22]. Bacteria with confirmed biodegradation capabilities belong mainly to *Proteobacteria* (α, β and γ), *Actinobacteria,* and *Bacilli*. Since the time when the first MC-degrading bacteria were described [23], several new bacterial strains able to degrade these toxins have been isolated from various environmental habitats worldwide [19,24,25,26,27]. The MC biodegradation is mostly based on a gene cluster containing the genes mlrA, mlrB, mlrC, and mlrD, and/or enzymatic processes [8]. Different MC-degrading products, which are results of the metabolism of MC-degrading bacteria, have been revealed. For example, linearized MC-LR, tetrapeptide, and Adda are three of the best-known MC-LR degradation products [25], whereas a recent study on *Sphingopxyis* sp. [28] revealed eight new different intermediate MC-LR-degradation products belonging to tripeptides (Adda-Glu-Mdha, Glu-Mdha-Ala, and Leu-MeAsp-Arg), dipeptides (Glu-Mdha, Mdha-Ala, and MeAsp-Arg), and amino acids (Leu and Arg). In general, *Sphingopxyis* sp. strain USTB-05 was able to completely degrade MC-LR, MC-RR, and MC-YR, and different products were observed [29,30,31,32]. Interestingly, it was shown that bacterial communities were also able to degrade oligopeptides other than MCs such as aerucyclamides and cyanopeptolins [33].

Macrophytes can biotransform MCs into less toxic compounds, which may be stored in vacuoles of storage cells [34]. MCs may be accumulated by plants, mainly directly, via diffusion or root adsorption of dissolved toxins. MCs may be biotransformed by the enzymatic transformation of glutathione and cysteine into MC conjugates via soluble glutathione transferases, or GST in plant cells. The conjugates are then broken down into cysteine conjugates, strengthening the cell’s internal transport and excretion of the conjugated toxins from the plant. Detoxication processes, which use GST, were described for different aquatic organisms ranging from invertebrates to fish [11,34,35,36,37]. *Spirodela polyrhiza* (a duckweed) is a member of the smallest aquatic free-floating plant family, Lemnaceae [38]. *S. polyrhiza* is larger than other duckweeds, with fronds of up to 1 cm in comparison to only a few mm for most other duckweed species. *S. polyrhiza* has several roots, whereas *Lemna gibba* and *L. minor* have only one root. *S. polyrhiza* occurs in different freshwater bodies. Members of duckweeds are known to have their submerged parts colonized by different microorganisms [39,40]. Duckweeds are used worldwide as a model plant organism [41]. Therefore, this species was chosen for the current study.

Most papers on MC biodegradation are based on studies carried out on bacterial strains isolated from natural aquatic environments or bacterial communities present in natural waters collected from waterbodies with previous cyanobacterial bloom events [8,19]. Less attention has been paid to other organisms [35,36]. Therefore, in this study, it was hypothesized that (1) MCs may be degraded in the presence of the floating aquatic plant *Spirodela polyrhiza* and microorganisms that develop and inhabit plants under laboratory conditions and (2) the biodegradation rate differs for three different MC variants: MC-RR, MC-LR, and MC-LF. The aims of this study were to: (i) identify microorganisms and enumerate their abundance in water collected from controls and variants containing macrophytes; (ii) detect MCs and their biodegradation products in water after incubation with *S. polyrhiza* and associated microorganisms; and (iii) study the accumulation of MCs and their biodegradation products in the duckweed exposed to MC-RR, MC-LR, and MC-LF.

## 2. Materials and Methods

### 2.1. Organisms

*S. polyrhiza* turions (the dormant stage of the plant) used in the experiments were obtained from the Duckweed Toxkit F test (MicroBio Tests, Belgium). Turions were germinated according to the instructions of the producer. They were incubated for three days in a Petri dish containing the “Steinberg medium” [42] and placed in a cabinet at 25 °C with continuous top illumination (6000 lux). *S. polyrhiza* was then cultured for three weeks in the standard medium in a 1 L flask in non-sterile room conditions (a natural 12:12 h light:dark cycle, temperature ca. 20 °C) to allow for the development of microorganisms in the medium and on plants. To imitate natural conditions, the underside of the fronds and roots were protected against the light with an aluminum foil, which was wrapped on the flask at the height of the plants remaining in a compact clump.

### 2.2. Biodegradation Assays

Separate tests were carried out for three MC variants (MC-RR, MC-LR, and MC-LF, Enzo Life Science, Lausen, Switzerland). Tap water sourced from underground intakes and filtered through a bacteriological filter (0.2 µm; Millipore, Cork, Ireland) was used as a medium both in controls and experimental variants containing *S. polyrhiza*. The tests were performed in three replicates in multi-well plates. MC-RR, MC-LR, and MC-LF were added respectively to each plate-well containing 2 mL of tap water to obtain the final concentration equal to 1000 ng/mL for each MC. The final concentration was controlled by HPLC analysis. Each three-week-old plant used in the experiments was randomly picked from the flask, gently rinsed three times with the experimental tap water, and placed into plates (one plant per one plate-well) containing MC-RR, MC-LR, and MC-LF, respectively. The plates were incubated in the cabinet for the following four and nine days at 20 °C with a light:dark cycle of 12:12 h (6000 lux). Controls with MCs and the tap water were set up in three replicates in a separate plate and incubated in the conditions described above.

### 2.3. Parameters Measured

After four and nine days of incubation, the total biomass (as fresh weight, FW; expressed to an accuracy of 0.1 mg) of each plant removed from each well of plates was weighed after rinsing two times with 50 mL of tap water and gently drying with filter paper. Then plants were frozen (at −20 °C) until the day of extraction for analysis of toxins. After nine days of the incubation period, 200 µL of water from each well was collected for microorganism identification and counting. The samples were immediately fixed with Lugol’s solution and formalin: glycerine mixture (3:1). The samples of experimental water, both after four and nine days of incubation, were placed in Eppendorf tubes, and after centrifugation (14,000× *g* for 10 min. at 17 °C), the supernatants were collected and the MC-degradation was stopped by freezing (at −20 °C). The supernatants were then analyzed by HPLC-PDA to determine the concentration of MCs and to detect and determine the concentration of the MC degradation products (expressed as equivalent of MC-RR, MC-LR, and MC-LF, respectively) as described below. The MCs degradation rate was calculated by the equation of (Cs-Cd)/Cs × 100%
where:C is the concentration of particular MCsCs is the concentration of particular MCs at the beginning of the experimentsCd is the concentration of particular MCs after four and nine days of incubation

### 2.4. Microorganism Identification and Counting

Algae found in water samples after the 9-day experiment were identified mostly to the genus level in accordance with professional manuals [43,44]. The abundance of bacterial and algal cells was estimated by direct microscopic counting (Zeiss Primo Star, Monument, PA, USA) of cells in the Bürker chamber (Merck Eurolab, Stockholm, Sweden). This is a simple manual method for enumeration of average cell number by counting cells in squares; each square for bacteria counting has an area of 1/400 mm^2^ and sample volume of 1/4,000,000 mL, whereas for algal cells, the area was 1/25 mm^2^ and the sample volume was 1/250,000 mL. For each sample, counting was performed in triplicates and in 20 squares of the counting grid each time.

### 2.5. Plant Extraction for MC Analysis

Plants incubated for four and nine days with MC-RR, MC-LR, and MC-LF, respectively, were ground with a glass spatula and then sonicated in 1.5 mL of 75% (*v/v*) methanol (gradient-grade, Merck, Frankfurt, Germany) for 5 min (50 W, ultrasonic homogenizer Sonopuls, Bandelin, Berlin, Germany). After centrifugation (14,000× *g* for 10 min. at 17 °C), the supernatants were collected and analyzed for MCs.

### 2.6. HPLC-PDA Analysis of Microcystins and Their Biodegradation Products in Water and Plants

The HPLC-photodiode array detection system (Shimadzu, Kyoto, Japan) was used for the detection and identification of MCs and their degradation products. The UV detection range was 200–300 nm. MC-RR, MC-LR, and MC-LF (Enzo Life Science, Lausen, Switzerland) were used as the standards. Separation of supernatant components was performed using a Purosphere column (125 × 3 mm, dp 5 µm, Merck, Darmstadt, Germany) with the mobile phase composed of acetonitrile (Merck, Burlington, VT, USA) and gradient-grade water for HPLC acidified with 0.05% trifluoroacetic acid (gradient 30–100%), at a flow rate 0.7 mL/min. MCs and their biodegradation products were identified on the basis of specific spectra and the time of elution compared with the spectra and elution times of standards. Each fresh MC standard used in the experiment, to prepare a calibration curve, and to identify MCs and their degradation products had one specific peak and spectrum.

### 2.7. Data Analysis

Normality distribution (Kolmogorov–Smirnov test) and homogeneity of variance (Levene’s test) were tested for all the data obtained from experiments to choose appropriate statistical tests. Significant differences among experimental variants were evaluated using one-factor analysis of variance (ANOVA). Pair-wise comparison of means was performed using the Tukey test (*p* < 0.05). All statistical tests were carried out using the Statsoft Statistica package v. 10 for Windows at a significance level of *p* < 0.05. All the data obtained in the tests were expressed as mean values (*n* = 3) ± standard deviation (SD).

## 3. Results

### 3.1. Microorganisms Assocciated with S. polyrhiza

The obtained results revealed that after the 9-day incubation period, different microorganisms developed in the experimental controls and variants containing *S. polyrhiza* (Figure 1). Bacteria occurred in all controls and variants with plants (Figure 1a), whereas green algae (except for *Kirchneriella* sp.) and diatoms were found only in the variants with the duckweed (Figure 1b–e). The estimated abundance of bacterial cells was lower in the controls than in the variants with plants (Figure 1a). The abundance of bacterial cells reached the lowest values in the controls with MC-RR (3.6 × 10^6^ ind./mL) and had almost five-fold higher values (17.1 × 10^6^ ind./mL) in the variants with the duckweed and MC-LF. In all controls, the abundance of bacterial cells was similar, and no statistically significant differences were found. However, in general, significant differences (*p* < 0.05) in abundance were found among the controls and variants with macrophytes. The statistically significant difference in abundance of bacterial cells was also found between the variants with *S. polyrhiza* and MC-RR and MC-LR, and *S. polyrhiza* and MC-LF (Figure 1b). *Chlorella* sp., flagellate green algae, and *Navicula* sp. (diatoms) were found only in variants with the duckweed, whereas *Kirchneriella* sp. was present in one control containing MC-RR (Figure 1b–e). In general, the abundance of these eukaryotes was between 1000 and 10,000 times lower than the abundance of bacterial cells. *Chlorella* sp. was the most abundant among eukaryotic microorganisms, and reached up to 7.5 × 10^3^ ind./mL (Figure 1b). The statistically significant differences in the abundance of *Chlorella* sp. were found between all controls and variants with *S. polyrhiza*, and between the variants with *S. polyrhiza* and microcystins RR and LR and *S. polyrhiza* and MC-LF. In the last variant containing MC-LF, the abundance of *Chlorella* sp. was the lowest (1.91 × 10^3^ ind./mL). The abundance of flagellate green algae (0.28–0.56 × 10^3^ ind./mL) found in the variants with *S. polyrhiza* and MC-LR and MC-LF (Figure 1c) was a few times lower than the abundance of *Navicula* sp. (Figure 1d), and it was similar to the abundance of *Kirchneriella* sp. (Figure 1e). The abundance of *Navicula* sp. was the lowest in the variants with MC-LF and the highest in the variants with MC-RR. However, the differences were not statistically significant (Figure 1d).

### 3.2. Biodegradation of MCs after the Incubation with S. polyrhiza and Associated Microorganisms

The study showed that MCs’ concentrations decreased after the incubation with *S. polyrhiza* and associated microorganisms (Figure 2), and the degradation rate was different for MC-RR, MC-LR, and MC-LF (Table 1). The highest statistically significant decrease in MC concentration (Figure 2) and the highest degradation rate (Table 1) were observed for MC-RR (followed by MC-LR) after nine days of their incubation with *S. polyrhiza* and associated microorganisms. For instance, after nine days, the MC-RR concentration decreased by 61% in comparison with the beginning concentration, and it was equal to 384.3 ng/mL (Table 1, Figure 1a). The degradation rate of MC-LR after nine days of incubation was 21% (Table 1), and the MC-LR concentration decreased to 886.0 ng/mL (Figure 1b). The average MC-LF concentration decreased from 850 ng/mL at the beginning of the experiment to 786.7 ng/mL at the experimental endpoint (Figure 1c); however, the changes observed were not statistically significant. There were no statistically significant differences among concentrations of particular MCs in the controls both after four and nine days of experiments.

Degradation products of MC-RR, MC-LR, and MC-LF, marked with capital letters A-F, were detected in water from all experimental variants containing *S. polyrhiza* and microorganisms after both four and nine days of incubation. In the controls, only A-LF and C-LF degradation products were observed after nine days of exposure (Figure 3, Table 2). In total, six, two, and four degradation products of MC decomposition were observed by HPLC for MC-RR, MC-LR, and MC-LF, respectively (Figure 3, Table 2). These products had the UV spectrum typical for the standard spectra and their retention times ranged from 5.6 to 16.2 min for MC-RR products (A-RR, B-RR, C-RR, D-RR, E-RR, and F-RR), from 19.3 to 20.2 min for MC-LR products (A-LR and B-RR), and from 28.9 to 33.0 min for MC-LF products (A-LF, B-LF, C-LF, and D-LF) (Table 1, Figure 4, Figure 5 and Figure 6).

Among six MC-RR degradation products detected, the D-RR product with the retention time of 10.6 min (Table 1, Figure 4) reached the highest concentration (Table 2). This product was detected after both four and nine days of incubation in the variants with *S. plyrhiza* and microorganisms. Its concentration increased during the experimental period and ranged between 331.1 and 379.5 ng equiv. of MC-RR/mL after nine days of incubation (Table 2). The F-RR product with the retention time of 16.2 min (Table 1, Figure 4) was the second product with the highest concentration observed and it was detected after four and nine days of exposure. Its concentration (16.5–58.3 ng equiv. of MC-RR/mL), as in the case of D-RR product, was the highest at the endpoint of the experiment (Table 2). The products A-RR, B-RR, C-RR, and E-RR were found in single samples and only after nine days of incubation. Their concentration did not exceed 20 ng equiv. of MC-RR/mL. Two MC-LR degradation products were found; the A-LR product occurred after nine days of incubation, whereas the B-LR product was detected after both four and nine days of the incubation period and reached from four to eight times higher concentrations than the A-LR product. Retention times of both products were longer than the retention time of MC-LR (Table 1, Figure 5). Four degradation products of MC-LF were found, but only the A-LF product with the retention time of 28.9 min was detected in all experimental variants with *S. polyriza* and microorganisms (Table 2), as well as in the 9-day control (Table 1, Figure 6). The concentration of this product (from 5.1 to 5.9 ng equiv. MC-LF/mL) was the highest after nine days of incubation in the variants with *S. polyrhiza* and microorganisms (Table 2). Products A-LF, C-LF, and D-LF were found in single samples, and it seems that their concentrations did not depend on the longevity of the incubation period. The analysis of the total concentrations of individual MCs and their degradation products, expressed as equivalent of MC, showed statistically significant differences between variants with plants and controls after four and nine days of incubation for MC-RR only (Table 2). No such statistically significant differences were found for the total concentration of both MC-LR and MC-LF and their degradation products.

No accumulation of MCs and their biodegradation products in the duckweed was observed after four or nine days of incubation with toxins.

## 4. Discussion

The present study is the first to demonstrate the phenomenon of the degradation of the hepatotoxic MC-RR, MC-LR, and MC-LF in the presence of macrophytes and the associated microorganisms which develop and inhabit plants under laboratory conditions. Since no accumulation of MCs and their biodegradation products in the duckweed *S. polyrhiza* was observed, it seems that microorganisms associated with plants could be responsible for the MC degradation observed. Tap water, poor in nutrients and used as a medium, could inhibit the macrophyte metabolism and toxin uptake [45], but it should not lower the ability of microorganisms to decompose MCs, since some bacterial strains, such as *Sphingomonas* 7CY, were able to degrade MC-RR independently from the presence of carbon and nitrogen sources [46]. Moreover, it has been shown that carbon and nitrates may have intricate roles in the MC biodegradation process ([22] and references therein).

Although the microbial communities of the rhizoplane and fronds of aquatic plants are poorly recognized [39,40,47], it was shown that submerged parts of *S. polyrhiza* exhibit a variety of microorganisms such as bacteria, cyanobacteria, and diatoms throughout their development [39], which is also suggested by the current study, which showed variability in abundance of particular microbial groups found in experiments. Heavy colonization by the microorganisms was observed at maturity, especially in the fully developed abaxial fronds and root caps [40,47]. Matsuzawa and co-authors [40] analyzed microbial communities in the rhizoplane of *S. polyrrhiza* during cultivation-dependent and independent analyses. The cultivation-based analysis revealed that the rhizoplane isolates were affiliated with *Alphaproteobacteria, Betaproteobacteria, Bacteroidetes,* and *Verrucomicrobia*. Interestingly, the results suggested that the rhizoplane of *S. polyrrhiza* forms a specific habitat for the bacteria within *Verrucomicrobia* phylum. Culture-independent analyses revealed eight bacterial classes (*Alphaproteobacteria, Betaproteobacteria, Gammaproteobacteria,* and *Deltaproteobacteria*) and phyla (*Bacteroidetes, Verrucomicrobia, and Planctomycetes*) and Cyanobacteria. The findings suggested that the microbes in the duckweed rhizoplane are comprised of a diverse array of readily cultured organisms. The recent study by Iwashita and co-authors [47] carried out on *S. polyrhiza, Lemna minor*, and *L. aequinoctialis* collected from freshwater environments, showed that among the 11 bacterial phyla inhabiting duckweeds, the phylum *Proteobacteria,* with confirmed MC-biodegradation capabilities [23], was the most predominant group in fronds (57.3–62.4%), roots (48.1–59.6%), and pond water (43.3%) [47]. However, the other constituents between plant samples and the pond water sample differed; seven and nine phyla, except for *Proteobacteria*, were detected in the root and frond samples, respectively, while only four phyla were found in the pond water. Rarely cultivated bacterial groups such as *Armatimonadetes* and *Verrucomicrobia* were found as well.

In the present investigation, bacteria, with abundance that was much higher than the abundance of other microorganisms found in the experimental conditions, are the best-known and most intensively studied organisms able to degrade MCs [19,24,25,26,27,48,49]. Numerous bacterial strains isolated from many different aquatic environments worldwide have strong MC-degrading abilities, and therefore, they may play a significant role in the natural degradation and removal of MCs. The majority of these bacteria belong to the genus *Sphingomonas*; however, other species from the genera *Acinetobacter*, *Arthrobacter, Bacillus, Novosphingobium, Paucibacter, Pseudomonas, Sphingopyxis*, and *Stenotrophomonas* are also known for MC biodegradation [8]. In the present study, at the high abundance of bacterial cells and the lower abundance of eukaryotic microorganisms, after nine days of incubation, the highest degradation rate (61%) was found for MC-RR, followed by MC-LR. In general, the degradation process was slower than in studies carried out on bacteria isolated from the natural environment [24,25,50,51]. For example, the recent study by Idroos and coauthors [51] showed that the bacterium *Stenotrophomonas maltophilia* strain 4B4 totally removed MC-LR within 10 days, while MC-RR and MC-LF were biodegraded within 12 and 14 days, respectively. In the current study, MC-LR was degraded by 21% after nine days, whereas the decrease in the concentration of MC-LF was not statistically significant. However, the MC degradation can be even faster, as the strain *Bacillus* sp. AMRI-03, isolated from a Saudi eutrophic lake affected by toxic cyanobacterial blooms, completely degraded MC-RR within five days (after a lag period of two days) [25], and different *Arthrobacter* sp. strains removed MC-LR within 72 h [24,50]. The bacterium *Novosphingobium* sp. KKU-25s, discovered recently as a new strain that is able to perform MC degradation [48], decomposed [Dha(7)]MC-LR within 24 h, whereas a novel strain YF1 of the genus *Sphingopyxis*, isolated from the eutrophic and Lake Taihu affected by cyanobacterial blooms, decomposed MC-LR within 120 min [49]. Najera and co-authors [52] showed that from a total of 24 bacterial strains isolated during the cyanobacteria blooming period from Jeziorsko Reservoir in Poland, only one strain JEZ-8L (belonging to the *Sphingosinicella* genus) that included the full mlrABCD gene cluster was found to completely degrade MC-LR. Combinations of two or more bacterial strains have also shown to be capable of degrading MC-LR. For example, seven isolates (*Acinetobacter* sp., *Hyphomicrobium aestuarii*, *Pseudoxanthomonas* sp., *Rhizobium* sp., *Sphingobium* sp., *Sphingomonas* sp., and *Steroidobacter* sp.) from a Taiwan reservoir in China [53] and ten isolates (*Acinetobacter* sp., *Aeromonas* sp., *Novosphingobium* sp., *Ochrobactrum* sp., *Pseudomonas* sp., *Rhodococcus* sp., *Sphingomonas* sp., *Sphingopyxis* sp., *Stenotrophomonas* sp., and *Steroidobacter* sp.) from a drinking water reservoir in Southern California [54] completely degraded MC-LR.

Both identified and unidentified bacterial communities isolated from natural waters affected by cyanobacterial mass development or blooms were able to entirely remove MC-LR (however, at different times) [19,55]. Edwards and co-authors [56] showed that the degradation rates of MC-LR, MC-LF, and nodularin in natural water samples differed and ranged from a half-life of four to eighteen days. The degradation processes were faster in the habitats suffering frequent HCBs and slower with a lag period in those without MC-exposure history. Dziga and co-authors [19] confirmed these findings for dmMC-LR and samples collected in 21 water bodies with previous cyanobacterial history. dmMC-LR incubated in 18 samples collected from these freshwater areas was degraded within 6 to 12 days. This may suggest a possible diversity of microorganisms involved in the MC-biodegradation process. The current study seems to confirm the previous findings, since although the abundance of bacterial cells in the controls and variants with *S. polyrhiza* differed only twice, the biodegradation was noted only in the variants with plants at simultaneous lack of bioaccumulation of MCs and their degradation products in the duckweed. This may suggest that in the controls and variants containing the duckweed, different bacterial species/strains developed; however, green algae found in the variants containing plants could also take a part in MC biodegradation. The current results clearly show that the MC-biodegradation processes (probably enzymatic processes and degradation pathways) may vary among different MC variants. Different degradation rates of the same MC variants observed in the present and above-mentioned studies might result from the variation in bacterial species or strains and the differences in microorganisms’ abundance. Generally, environmental factors such as temperature, pH, and nutrient and metal ions concentrations, which may stimulate or inhibit bacterial growth, may influence both aerobic and anaerobic MC-biodegradation efficiency [22,51].

In the present study, green-algae *Chlorella* sp., flagellate green algae, and diatoms *Navicula* sp., which occurred in the experimental variants containing the duckweed, could take a part in the MC biodegradation; however, the knowledge about the ability of algae, including flagellates, to decompose MCs is very limited [22,57]. For example, the flagellate *Diphylleia rotans* degraded MCs under axenic conditions [57]. The mixotrophic chrysophyte *Poterioochromonas* sp. ZX1 (*Ochromonas*) degraded 82% of intracellular and extracellular MC-LR (a total of 114 ng/mL) while digesting 7.3 × 10^5^~4.3 × 10^6^
*Microcystis* cells/mL [58,59] and the process was boosted by higher temperature [30]. It is important to remember that the natural MC-biodegradation process always involves co-existing organism systems (e.g., algae–bacteria or protists–bacteria) [60,61]. It is well-known that microorganisms associated with duckweeds may contribute to the purification of water in the environment [39,40]; therefore, they could be also useful in purification of water from MCs.

In the current study, six, two, and four biodegradation products for MC-RR, MC-LR, and MC-LF, respectively, were detected after nine days of MC incubation with *S. polyrhiza* and the associated microorganisms. Different intermediate degradation products of MC-LR, MC-RR, and MC-YR were also detected during the studies on the bacterial strain *Sphingopxyis* sp. [28,29,31,32]. For instance, a strain USTB-05, isolated from Lake Dianchi in China, degraded MC-RR at the rate of 16.7 µg/mL per day [29,62], and the analysis of the enzymatic degradation pathways for MC-RR showed that Adda-Arg peptide bond of MC-RR was cleaved, and then a hydrogen and a hydroxyl were combined onto the NH_2_ group of Adda and the carboxyl group of arginine to form a linear molecule. This intermediate product was formed within the first few hours. Then, after 24 h, the final product (a linear MC-RR with two small peptide rings) was formed through a dehydration reaction. The studies showed that MC-RR may undergo different transformations, and different products were formed by various bacteria in natural lakes and reservoirs, which seems to be in agreement with the presented study. In the current study, only two degradation products of MC-LR were detected by HPLC, whereas, besides three well-known MC-LR degradation products such as linearized MC-LR, tetrapeptide, and Adda, eight new different intermediate degradation products were discovered [28]. Three tripeptides (Adda-Glu-Mdha, Glu-Mdha-Ala, and Leu-MeAsp-Arg), three dipeptides (Glu-Mdha, Mdha-Ala, and MeAsp-Arg) and two amino acids (Leu and Arg) were identified during studies on the bacterium *Sphingopxyis* sp. Dziga and co-authors [19] reported four main products of dmMC-LR degradation after exposure to water from 21 freshwater habitats suffering from HCBs. The two main products were cyclic dmMC-LR with modifications in the Arg-Asp-Leu region and the tetrapeptide Adda-Glu-Mdha-Ala. Since, in the current study, retention times of MC-LR products were longer than the retention time of the MC-LR standard, they might be still cyclic compounds [19]. Edwards and co-authors [56] found novel intermediate degradation products of MC-LF which included demethylation, hydrolysis, decarboxylation, and condensation of the parent compounds.

No accumulation of MCs and their degradation products in *S. polyrhiza* was observed during the present study. However, the phenomenon of MC accumulation in plants is well-known. For instance, Romero-Oliva and co-authors [34] showed an accumulation of MCs in four macrophytes (*Polygonum portoricensis, Eichhornia crassipes, Typha* sp., and *Hydrilla verticillata*) coexisting with the blooms of coccoid cyanobacterium *Microcystis aeruginosa*. A laboratory study showed that 30% of the total (90 ng/mL) MCs were taken up by *H. verticillata* within just seven days, and plants accumulated more MC-LR than MC-RR and MC-YR. The accumulation of MCs in plants depends on the route of exposure, the dose and kind of toxin, target organs, the duration of exposure, the target tissues, and specific species [63]. For example, the duckweed *L. minor* accumulated MCs up to a concentration of 0.3 ng/mg wet weight after 5-day exposure to MCs at the concentration of 20,000 ng/mL with an accumulation rate of 58 ng g/day [64]. In the present study, *S. polyrhiza* was exposed to the concentration of 1000 ng/mL; however, the metabolism of plants could be affected by a lack of appropriate nutrient concentrations. Interestingly, MC removal by aquatic plants has been well-documented, but most studies conducted this process under non-aseptic conditions. Therefore, there was insufficient evidence to prove the biodegradation capability of plants or to reject the possibility that microbes attached to plants participate in the MC degradation [22]. Nevertheless, the 7-day study by Isobe and co-authors [65] carried out on the sterile hydroculture *of Portulaca oleracea* cv. and on a germ-free solid medium showed that MC-LR was metabolized by the plants in 24%, 54%, and 100% of the initial concentrations of 2000, 200, and 20 ng/mL, respectively. MC-LR concentrations of 20 and 200 ng/mL were quickly degraded by the plants into unknown compound(s) of lower toxicity. This suggests that the concentrations of MCs may be an important factor in the efficiency of the MC-degradation process in plants. Interestingly, the statistical analysis of the changes in the total concentrations of individual MCs and their degradation products, which revealed significant differences for MC-RR only, suggests that the MC biodegradation was a main contributor to the decrease in MC concentration; however, the MC adsorption onto the duckweed fronds and/or roots cannot be completely excluded and requires further studies.

Interestingly, the current study revealed different toxicity of particular MCs and/or mixtures of MCs and their degradation products to bacteria and eukaryotic microorganisms, which is in agreement with previous discoveries ([66] and references therein). For instance, the experiments suggest that among the studied MC variants, MC-RR was the most toxic to bacteria, followed by MC-LR and MC-LF, since the number of bacterial cells was the lowest in the variants containing MC-RR and the highest in the variants with MC-LF. However, MC-LF showed the highest toxicity to *Chlorella* sp. and *Navicula* sp. MC-LR and MC-RR seemed to be less toxic. In general, the toxicity of different MC variants varied substantially when administered by intraperitoneal injection (i.p.) to mice [13,66], and MC-LR is one of the most toxic variants (LC_50_ = 50 µg/kg) [64]. The replacement of the hydrophobic leucine (L) in position X with a more hydrophilic amino acid (e.g., arginine, R) reduces MC-LR toxicity [13,66]. The study by Isobe and co-authors [65] showed that some unknown compound(s) of MC-LR biodegradation had lower toxicity than pure MC-LR monitored by protein phosphatase inhibition assay (PPIA). No mouse toxicity data are available for MC-LF, but *in vitro* studies showed that MC-LF and MC-LW, in which the hydrophobic amino acids tryptophan (W) or phenylalanine (F) occupy position Y, were distinctly more toxic to human hepatocytes, organic anion transporting polypeptide (OATP)-transfected embryonic kidney cells, and Caco-2 cells than MC-LR [67,68]. MC-LF was also more toxic to the protozoan *Tetrahymena pyriformis* than MC-LR [69]. The present study confirms that MC toxicity may be species-specific.

## 5. Conclusions

The presented study is the first that demonstrates the phenomenon of the degradation of the hepatotoxic cyanobacterial toxins MC-RR, MC-LR, and MC-LF in the presence of macrophytes (the duckweed *S. polyrhiza*) and the associated microorganisms which developed under laboratory conditions. As no accumulation of MCs and their degradation products in *S. polyrhiza* was found, it seems that microorganisms such as bacteria and eukaryotic algae (e.g., *Chlorella* sp. and *Navicula* sp.) might take a part in the MC-biodegradation processes. Bacterial biodegradation of microcystins has been intensively investigated in recent years; however, studies on MC-degraders belonging to eukaryotes are very limited. Particularly, the studies on algae as potential MC degraders are missing. This work shows that further research in this field is strongly required, since it may provide important issues for the broadening of the knowledge about MC degradation and for the practical application of MC biodegradation in water treatment and water management.

## Figures and Tables

**Figure 1 ijerph-19-06086-f001:**
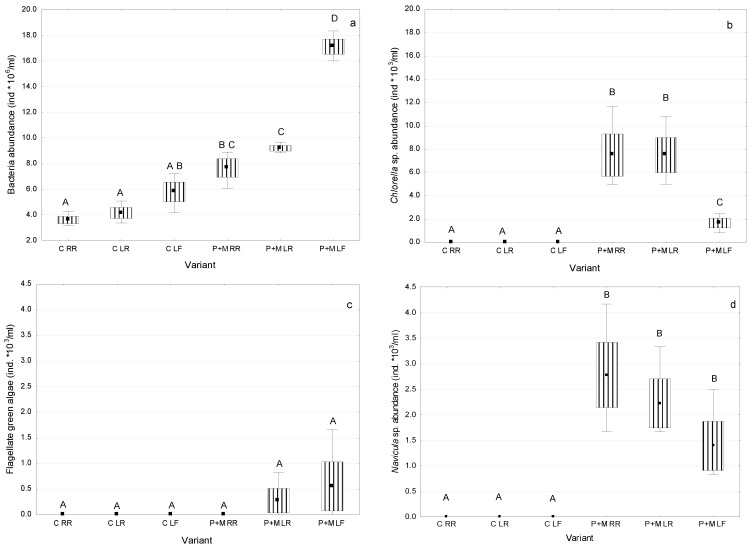
Abundance of bacterial cells (**a**), *Chlorella* sp. (**b**), flagellate green algae (**c**), *Navicula* sp. (**d**), and *Kirchneriella* sp. (**e**) in controls (C) and variants with *S. polyrhiza* (P + M) containing MCs after nine days of incubation (mean ± SD; *n* = 3). Uppercase letters (A, B, C, D) indicate statistically significant differences among experimental variants (ANOVA, Tukey’s test, *p* < 0.05). C—controls containing MCs, but without plants; P + M—experimental variants with MCs, plants, and associated microorganisms.

**Figure 2 ijerph-19-06086-f002:**
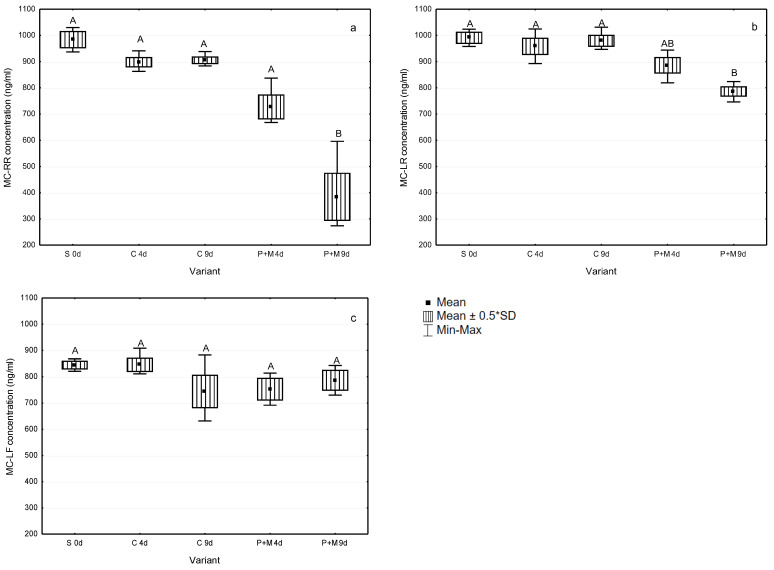
Concentrations of MC-RR (**a**), MC-LR (**b**), and MC-LF (**c**) in controls (C) and experimental variants with *S. polyrhiza* and associated microorganisms (P + M) after four and nine days of incubation (mean ± SD; *n* = 3). Uppercase letters (A, B) indicate statistically significant differences between experimental variants (ANOVA, Tukey’s test, *p* < 0.05). S—experimental starting point; C—controls containing MCs, but without plants; P + M—experimental variants with MCs, plants, and associated microorganisms.

**Figure 3 ijerph-19-06086-f003:**
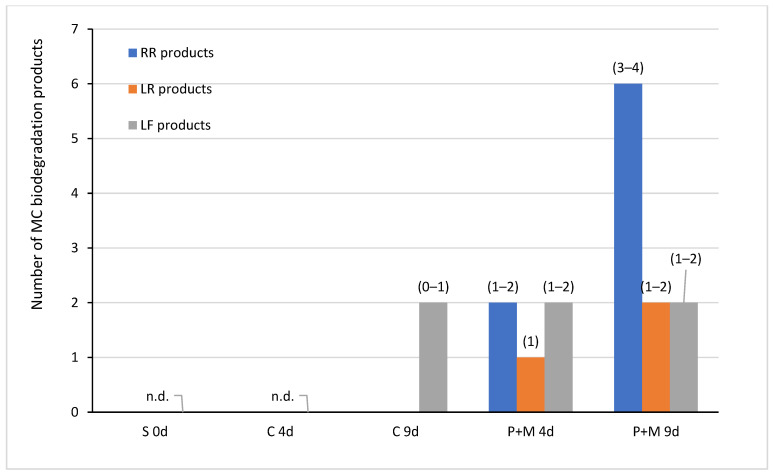
The total number of the MC biodegradation products in controls (C) and variants with *S. polyrhiza* and associated microorganisms (P + M) after four and nine days of the incubation period. In parenthesis, the number of products per sample is shown. S—experimental starting point; C—controls containing MCs, but without plants; P + M—experimental variants with MCs, plants, and associated microorganisms.

**Figure 4 ijerph-19-06086-f004:**
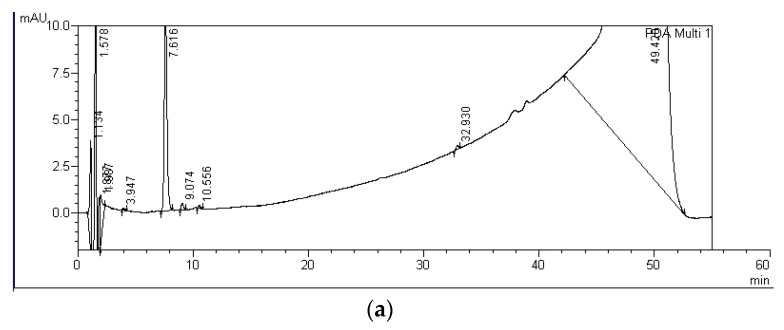
HPLC chromatograms indicating the progressive reduction peaks relevant to MC-RR and growing peaks of MC-RR degradation products after four (**a**) and nine days (**b**) of incubation with *S. polyrhiza* and associated microorganisms. Peaks and their retention times for MC-RR and its degradation products were as follows: MC-RR—7.716–7.761 min.; C-RR—8.580 min., D-RR—10.542–10.556 min.; E-RR—14.592, and F-RR—16.199 min.

**Figure 5 ijerph-19-06086-f005:**
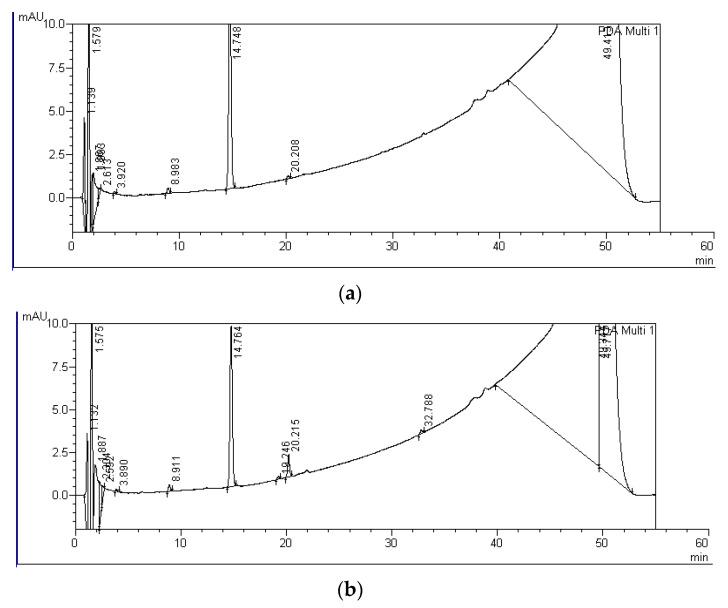
HPLC chromatograms indicating the progressive reduction peaks relevant to MC-LR and growing peaks of MC-LR degradation products after four (**a**) and nine days (**b**) of incubation with *S. polyrhiza* and associated microorganisms. Peaks and their retention times for MC-LR and its degradation products were as follows: MC-LR—14.748–14.764 min.; A-LR—19.246 min. and B-LR—20.215–20.208 min.

**Figure 6 ijerph-19-06086-f006:**
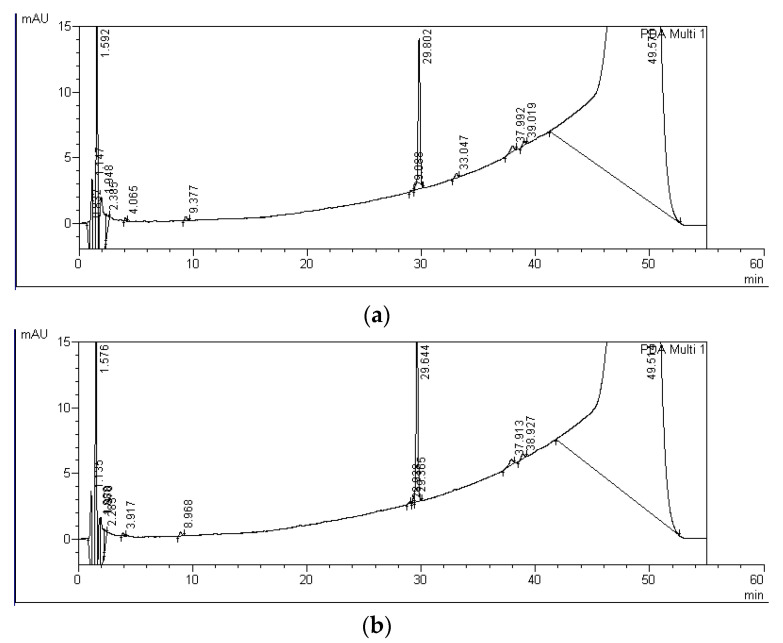
HPLC chromatograms indicating the peaks of MC-LF and its degradation product after four (**a**) and nine days (**b**) of incubation with *S. polyrhiza* and associated microorganisms. Peaks and their retention times for MC-LF and its degradation product were as follows: A-LF—28.938–29.088 min. and MC-LF—29.644–29.802 min.

**Table 1 ijerph-19-06086-t001:** Biodegradation of MC-RR, MC-LR, and MC-LF and the retention times of degradation products.

MC Variants	MCs Degradation Rates (%)		HPLC Detection of Degradation Products and Their Retention Times (Minutes)					
	4 d	9 d	A-*	B-*	C-*	D-*	E-	F-
MC-RR _C_	9 ± 4	8 ± 3	-	-	-	-	-	-
MC-LR _C_	3 ± 7	1 ± 5	-	-	-	-	-	-
MC-LF _C_	0 ± 6	11 ± 15		-	C-LF: 30.0	D-LF: 33.0	-	-
MC-RR _P+M_	26 ± 10	61 ± 19	A-RR: 5.6	B-RR: 6.5	C-RR: 8.6	D-RR: 10.6	E-RR: 14.6	F-RR: 16.2
MC-LR _P+M_	11 ± 6	21 ± 4	A-LR: 19.3	B-LR: 20.2	-	-	-	-
MC-LF _P+M_	11 ± 10	7 ± 9	A-LF: 28.9	B-LF: 29.4	C-LF: 30.0	D-LF: 33.0	-	-

C—controls without plants, P + M—variants with plants and microorganisms; * for MC-RR, MC-LR, or MC-LF, respectively.

**Table 2 ijerph-19-06086-t002:** Minimal and maximal concentrations of residual MCs and biodegradation products of MC-RR, MC-LR, and MC-LF after four and nine days of incubation with *S. polyrhiza* and associated microorganisms. S—experimental starting point; C—controls containing MCs, but without plants; P + M—experimental variants with MCs, plants, and associated microorganisms.

MCs and Their Degradation Products	Units	Experimental Variants				
		S 0d	C 4d	C 9d	P + M 4d	P + M 9d
A-RR	ng equiv. MC-RR/mL					0–7.9
B-RR						0–5.5
MC-RR	ng MC-RR/mL	936.8–1030.2	862.9–941.06	883.6–938.2	667.9–836.8	273.4–595.7
C-RR	ng equiv. MC-RR/mL					0–16.0
D-RR					11.3–35.9	331.1–379.5
E-RR						0–20.0
F-RR					0–45.7	16.5–58.3
Total concentration	ng equiv. MC-RR/mL	936.8–1030.2	862.9–941.06	883.6–938.2	712.9–848.0 *	729.5–851.3 **
MC-LR	ng MC-LR/mL	958.5–1023.0	892.4–1024.7	947.0–1031.2	818.8–943.9	745.8–824.2
A-LR	ng equiv. MC-LR/mL					0–10.0
B-LR					0–27.8	37.0–80.4
Total concentration	ng equiv. MC-LR/mL	958.5–1030.2	892.4–1024.7	947.0–1031.2	846.6–950.17	825.3–862.4
A-LF	ng equiv. MC-LF/mL			0–5.2	0–7.0	5.1–5.9
B-LF						0–23.0
MC-LF	ng MC-LF/mL	820.9–868.5	811.6–908.4	631.4–882.6	691.7–813.9	730.1–843.4
C-LF	ng equiv. MC-LF/mL			0–6.6		
D-LF					0–30.6	
Total concentration	ng equiv. MC-LF/mL	820.9–868.5	811.6–908.4	636.6–882.6	697.0–851.6	735.1–872.3

C—controls containing MCs, but without plants, P + M—variants with plants and microorganisms. *—statistically significant differences in comparison with all controls (*p* < 0.05); **—statistically significant differences in comparison with the start controls (*p* < 0.05).

## Data Availability

Data are available upon request.

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
