# Peer review of "Degradation of Three Microcystin Variants in the Presence of the Macrophyte Spirodela polyrhiza and the Associated Microbial Communities"

_ijerph, 2022, doi:10.3390/ijerph19106086_

Round 1
Reviewer 1 Report
Please address the following comments:
1. Justification for using Spirodela polyrhiza should be provided in the Introduction.
2. Any MCs adsorbed onto the leaf and root of the plant?
3. Is there a control without the plant but with the bacterial and algae? I'm not convinced the role of plants played here.
Author Response
Thank you very much for the valuable comments and suggestions. All were taken into consideration. The detailed answers are below.
Please address the following comments:
- Justification for using Spirodela polyrhiza should be provided in the Introduction.
Response: Justification was added in the Introduction, lines 79-86.
Any MCs adsorbed onto the leaf and root of the plant?
Response: Adsorption of MCs onto the leaf and root of the plant was not studied. However, the statistical analysis of changes in the total concentrations of individual MCs and their degradation products, expressed as equivalent of MC, showed statistically significant differences between variants with plants and controls after four and nine days of incubation for MC-RR only. No such statistically significant differences were found for the total concentration of MC-LR and MC-LF and their degradation products (Table 2, lines 300-305). This may suggest that biodegradation was a leading process. However, the MC adsorption cannot be completely excluded as was mentioned in the discussion (496-500).
Is there a control without the plant but with the bacterial and algae? I'm not convinced the role of plants played here.
Response: The control without the plants but with microorganisms and tap water was not set up. The manuscript presents the preliminary studies. The obtained results contributed to undertaking further research on S. polyrhiza (both young and mature plants), as well as bacteria and algae without plants and MCs degradation. These experiments were carried out with the use of Steinberg medium, not tap water. Hence, in my opinion, the obtained results cannot be used as a control for the data presented in the manuscript. Nevertheless, the studies confirmed that the degradation process was carried out by microorganisms associated with S. polyrhiza and will be published after obtaining more results such as the identification of bacteria and MC-degradation products.
Reviewer 2 Report
This study presents the result of Degradation of three microcystin variants in the presence of the macrophyte Spirodela polyrhiza and the associated microbial communities. It is found that MC-RR, MC-LR and MC-LF were degraded in varying degrees in the presence of S. polyrhiza and associated microorganisms that developed under laboratory conditions. The manuscript is written well and the following queries should be addressed before considering further processing.
Abstract
Give the full form of MC-RR, and MC-LF
Introduction
Give a brief account on Spirodela polyrhiza
Methods
What is the rationale for fixing the concentration of MC as 1000 ng/ml?
Why was the bacterial count not done by Total viable count?
Results
The name of microbes should be in italics
What are the names of these degradation products of MC-RR, MC-LR and MC-LF?
Any identification studies have been conducted about these products?
Why is the microbial community involved in the degradation not studied?
Author Response
Reviewer 2
This study presents the result of Degradation of three microcystin variants in the presence of the macrophyte Spirodela polyrhiza and the associated microbial communities. It is found that MC-RR, MC-LR and MC-LF were degraded in varying degrees in the presence of S. polyrhiza and associated microorganisms that developed under laboratory conditions. The manuscript is written well and the following queries should be addressed before considering further processing.
Thank you very much for the opinion and valuable comments and suggestions. All were taken into consideration. The detailed answers are below.
Abstract
Give the full form of MC-RR, and MC-LF
Response: Done - lines 13-14
Introduction
Give a brief account on Spirodela polyrhiza
Response: A brief account on S. polyrhiza was added in the Introduction, lines 79-86.
Methods
What is the rationale for fixing the concentration of MC as 1000 ng/ml?
Response: The concentration of MCs equal 1000 ng/ml was fixed as this concentration is commonly used in laboratory studies on MC biodegradation (for instance Jones et al. 1994, Dziga et al. 2017) and helps to compare results from different studies.
Why was the bacterial count not done by Total viable count?
Response: The limited volume of samples made it necessary to count all organisms together, using a microscope and the Burker chamber. Hence, no specific filtration and fluorescence method was used to count bacteria and they were not differentiated into dead and alive. Therefore, only the total number of all bacterial cells was estimated.
Results
The name of microbes should be in italics
Response: I am sorry for these mistakes and oversight. The names were corrected.
What are the names of these degradation products of MC-RR, MC-LR, and MC-LF?
Response: The letter designations of names of unidentified degradation products were taken from the work Dziga et al. 2017, that is cited in the manuscript. The full names were given – Table 1, lines 261-264.
Any identification studies have been conducted about these products?
Response: Identification of these products is planned in cooperation with another university, however, the date of the analysis is unknown.
Why is the microbial community involved in the degradation not studied?
Response: The paper presents preliminary studies on the biodegradation of MCs. A series of further experiments with S. polyrhiza (yang and mature plants) and without plants but with microorganisms were carried out. Presently, I am looking for a laboratory that will identify bacteria.
Reviewer 3 Report
- There are many improper scientific name writing. The scientific name should be italic.
- Method: not show the detail of the method.
2.1 what company of bacteriological filter did the author use?
2.2 how to count the bacteria? Ref 41 is incorrect because it is not correlated to the method at all. Then please check all references
2.3 how to calculate the kinetic rate?
2.4 how to calculate the degradation rate?
2.5 how to calculate the abandance? It is not clear.
- Result:
3.1 want to see one more control, which is P+M without the microcystins.
3.2 From the result, it seems that bacteria play more important role in degrading MC than eukaryotic. The authors should classify/identify bacteria instead of eukaryotes.
3.3 because the author didn’t show how to calculate the degradation rate. It is hard to understand if it is the accumulation of degradation or on a specific day (day 4 and 9)
- Discussion and conclusion:
The author should emphasize what is new in the work. Unfortunately, it hard to see new observations from the manuscript.
Author Response
Reviewer 3
Thank you very much for the valuable comments and suggestions. All were taken into consideration. The detailed answers are below.
There are many improper scientific name writing. The scientific name should be italic.
Response: I am sorry for these mistakes and oversight. They were corrected.
Method: not show the detail of the method.
Response: Methods were shown in more detail.
2.1 what company of bacteriological filter did the author use?
Response: The information was added – line 114
2.2 how to count the bacteria? Ref 41 is incorrect because it is not correlated to the method at all. Then please check all references
Response: The description of the counting method was improved (lines 148-153). The reference cited is not directly devoted to the method, but the authors describe and use the method to count bacterial cells. However, to avoid doubts and inaccuracies, the reference was removed from the manuscript.
2.3 how to calculate the kinetic rate?
Response: The kinetic rate was not calculated. Only the degradation rate was calculated and this was clarified and the information was added (lines 139-144). The title of the second column in Table 1 was corrected.
2.4 how to calculate the degradation rate?
Response: The information about the calculation of the degradation rate was added (lines 139-144)
2.5 how to calculate the abandance? It is not clear.
Response: The description of counting method was improved (lines 148-153)
Result:
3.1 want to see one more control, which is P+M without the microcystins.
Response: Unfortunately, the control P+M without microcystins was not set as the aim of the paper was to study the MCs degradation. The potential effect of microcystins on particular microorganisms is discussed as a side topic. I am asking for permission to leave this part of the work in its current form.
3.2 From the result, it seems that bacteria play more important role in degrading MC than eukaryotic. The authors should classify/identify bacteria instead of eukaryotes.
Response: The paper presents preliminary studies on the biodegradation of MCs. A series of further experiments with and without S. polyrhiza were carried out. Presently, I am looking for a laboratory that will identify bacteria.
3.3 because the author didn’t show how to calculate the degradation rate. It is hard to understand if it is the accumulation of degradation or on a specific day (day 4 and 9)
Response: The information about the calculation of the degradation rate was added (lines 139-144) and the title in the table was changed to clarify the information included.
Discussion and conclusion:
The author should emphasize what is new in the work. Unfortunately, it hard to see new observations from the manuscript.
Response: The new observations were highlighted in the discussion (lines 333-338) and conclusion (lines 523-536).
Reviewer 4 Report
This manuscript provides results of an interesting study indicating that associated microorganisms may be responsible for MC degradation. I have attached the PDF file with comments and requested corrections noted in the paper. In addition to the comments, I am requesting that the figures be further clarified. It is very difficult to follow the figures when S, C, etc. are not defined. Additionally, it would be helpful to clarify what control was used for each figure in the figure legend. I am also concerned that the degradation products were determined based on HPLC peaks and that there were no controls apparently used to determine the products. These peaks can shift depending on multiple factors. There should be further discussion explaining the certainty of each peak, and how you can be sure that it was a degradation product and not another compound coming off at that peak. Please clarify how you know with certainty that each peak is a degradation product. Please also state which peak is for which degradation product. The degradation products were initially mentioned in the discussion section. These should be explicitly states in the intro, results, figures, and discussion rather than just calling them product A, B, C, etc.

Author Response
Reviewer 4
This manuscript provides results of an interesting study indicating that associated microorganisms may be responsible for MC degradation. I have attached the PDF file with comments and requested corrections noted in the paper.
Thank you very much for the valuable comments and suggestions. All were taken into consideration. The detailed answers are below.
In addition to the comments, I am requesting that the figures be further clarified. It is very difficult to follow the figures when S, C, etc. are not defined. Additionally, it would be helpful to clarify what control was used for each figure in the figure legend.
Response: Figures’ captions were clarified according to the suggestion.
I am also concerned that the degradation products were determined based on HPLC peaks and that there were no controls apparently used to determine the products. These peaks can shift depending on multiple factors. There should be further discussion explaining the certainty of each peak, and how you can be sure that it was a degradation product and not another compound coming off at that peak. Please clarify how you know with certainty that each peak is a degradation product.
Response: To clarify the issue, an explanation was added in the methods section (Lines 174-176). Each fresh MC standard, that was used in the experiment, as well as to prepare a calibration curve, and to identify MCs and their degradation products, had one specific peak and spectrum. Identification of the degradation products needs a method that is different from HPLC (it is LC/MS-MS) and is planned in cooperation with another university, however, the date of the analysis is unknown. The peaks were identified as degradation products on the base of spectra that were characteristic or very similar to spectra that are characteristic for MC-RR, MC-LR and MC-LF standards. The elution time was also important. As separate experiments for MC-RR, MC-LR and MC-LF were set up, it is 100% sure that the products came from the degradation of a specific, one standard.
Please also state which peak is for which degradation product. The degradation products were initially mentioned in the discussion section. These should be explicitly states in the intro, results, figures, and discussion rather than just calling them products A, B, C, etc.
Response: The letter designations of names of the degradation products were taken from the work Dziga et al. 2017, which is cited in the manuscript. The corrections were made according to the suggestion (Figs. 4-6, Table 1, lines 260-263).
Sugestions marked in the text
Line 13 – corrected
Line 20 – added
Line 27 – changed
Line 28 – changed
Lines 13-14 and 34-37 – corrected
Line 54 – removed
Table 3 and Figures – defined
Line 335 – corrected. There should be “and”
Line 348 – corrected
Line 418 – changed to “differed twice”
Round 2
Reviewer 1 Report
My comments have been addressed.
Reviewer 3 Report
Thank you for all changes.
It would be nice to see the following paper on microbial community from duckweed.
Reviewer 4 Report
Thank you for making the revisions.